# Luteolin Improves Perivascular Adipose Tissue Profile and Vascular Dysfunction in Goto-Kakizaki Rats

**DOI:** 10.3390/ijms222413671

**Published:** 2021-12-20

**Authors:** Marcelo Queiroz, Adriana Leandro, Lara Azul, Artur Figueirinha, Raquel Seiça, Cristina M. Sena

**Affiliations:** 1Institute of Physiology, iCBR, Faculty of Medicine, University of Coimbra, 3000-548 Coimbra, Portugal; marceloflavioqueiroz@hotmail.com (M.Q.); adrianaleandro94@hotmail.com (A.L.); lara.r.s.azul@gmail.com (L.A.); rmfseica@gmail.com (R.S.); 2LAQV, REQUIMTE, Faculty of Farmacy, University of Coimbra, 3000-548 Coimbra, Portugal; amfigueirinha@gmail.com

**Keywords:** luteolin, type 2 diabetes, endothelial dysfunction, inflammation, oxidative stress

## Abstract

We investigated the effects of luteolin on metabolism, vascular reactivity, and perivascular adipose tissue (PVAT) in nonobese type 2 diabetes mellitus animal model, Goto-Kakizaki (GK) rats. Methods: Wistar and GK rats were divided in two groups: (1) control groups treated with vehicle; (2) groups treated with luteolin (10 mg/kg/day, for 2 months). Several metabolic parameters such as adiposity index, lipid profile, fasting glucose levels, glucose and insulin tolerance tests were determined. Endothelial function and contraction studies were performed in aortas with (PVAT+) or without (PVAT−) periaortic adipose tissue. We also studied vascular oxidative stress, glycation and assessed CRP, CCL2, and nitrotyrosine levels in PVAT. Results: Endothelial function was impaired in diabetic GK rats (47% (GK − PVAT) and 65% (GK + PVAT) inhibition of maximal endothelial dependent relaxation) and significantly improved by luteolin treatment (29% (GK − PVAT) and 22% (GK + PVAT) inhibition of maximal endothelial dependent relaxation, *p* < 0.01). Vascular oxidative stress and advanced glycation end-products’ levels were increased in aortic rings (~2-fold, *p* < 0.05) of diabetic rats and significantly improved by luteolin treatment (to levels not significantly different from controls). Periaortic adipose tissue anti-contractile action was significantly rescued with luteolin administration (*p* < 0.001). In addition, luteolin treatment significantly recovered proinflammatory and pro-oxidant PVAT phenotype, and improved systemic and metabolic parameters in GK rats. Conclusions: Luteolin ameliorates endothelial dysfunction in type 2 diabetes and exhibits therapeutic potential for the treatment of vascular complications associated with type 2 diabetes.

## 1. Introduction

Type 2 diabetes (T2D) is a complex multifactorial disease with major vascular complications and increased incidence of cardiovascular disease [1,2,3]. Endothelial dysfunction, an earlier pathophysiological event in the progression to atherosclerosis, exhibits an abnormal vascular reactivity due to impaired nitric oxide (NO) bioavailability. Endothelial dysfunction characteristics include reduced vasorelaxation, proinflammation, and prothrombotic properties [4]. In diabetes, metabolic dysfunction leads to damage in the vasculature involving increased oxidative stress, low-grade inflammation, endothelial dysfunction, and perivascular adipose tissue (PVAT) dysfunction [4,5,6].

Goto-Kakizaki (GK) rats are a nonobese diabetic model of T2D with similar characteristics to human diabetic disease (hyperglycemia, insulin resistance, vascular, and adipose tissue dysfunction) and suitable to evaluate pharmacological therapies capable of reducing the vascular burden associated with T2D [6]. GK rats were obtained by the selective breeding of glucose intolerant Wistar rats making them the natural nondiabetic controls.

The therapeutic potential of medicinal plants has an enormous importance for human health. Phenolic compounds, including flavonoid derivatives with anti-inflammatory and antioxidant properties [7] have been obtained from different natural plants. Human and animal studies have suggested that several flavonoid compounds with antioxidant properties could protect against oxidative stress and prevent obesity and related comorbidities [8].

Luteolin, 3′,4′,5,7-tetrahydroxyflavone, is a polyphenolic flavonoid, frequently present in some medicinal plants, vegetables, and fruits such as parsley, broccoli, celery, carrots, peppers, onion leaves, apple skins, and in spices [9]. Luteolin has anti-inflammatory, antioxidant, anticancer, and antimicrobial effects [9]. It was previously described that this flavone has beneficial effects on oxidative stress and inflammation associated with cardiovascular diseases [10,11,12,13]. In addition, luteolin hampers cholesterol biosynthesis [14], decreases insulin resistance and hepatic steatosis [15,16,17], and enhances endothelial NO synthase (eNOS) gene expression [18].

However, the impact of luteolin treatment in the regulation of vascular dysfunction and PVAT properties in T2D remains elusive. We hypothesized that luteolin might ameliorate endothelial dysfunction and reduce PVAT pro-inflammatory and vasoconstrictive profile in diabetic GK rats. The present study was carried out to evaluate the protective effects of luteolin on endothelial dysfunction and PVAT phenotype in GK rats and evaluate its influence on oxidative stress and inflammation in GK rats. We studied vascular function and characterized vascular oxidative stress, glycation, and the alterations observed in PVAT regarding their inflammatory and vasoconstriction profile. To our knowledge, this is the first time that the impact of luteolin treatment in vivo on endothelial function and its surrounding periaortic adipose tissue is studied in diabetic GK rats providing new insights to understand the mechanisms of action underlying this nutraceutical.

## 2. Results

### 2.1. Animal Characteristics

Food intake did not significantly change between W and WL groups and GK and GKL groups during the experimental period (Appendix A). GK rats exhibited significantly lower body weight and presented elevated non-HDL-cholesterol, triglycerides, and total cholesterol levels compared to age-matched Wistar rats (Table 1). Luteolin treatment significantly decreased body weight, adiposity index, and cholesterol levels in Wistar and GK rats (Table 1).

Glycemia, fasting, and 2 h after glucose load were elevated in diabetic GK rats compared with control Wistar rats (Figure 1A–C and Table 1). In an intraperitoneal glucose tolerance test (IPGTT), GK revealed marked glucose intolerance (Figure 1A,B). The glucose area under the curve (AUC) and fasting glycemia were significantly increased in GK rats when compared to Wistar rats (Figure 1B,C, Table 1). Luteolin significantly improved IPGTT and fasting glucose (Figure 1A,C) in diabetic GK rats. The glucose AUC was significantly reduced in luteolin groups (WL, GKL) when compared to the respective control groups (Figure 1B).

Insulin tolerance test (ITT) revealed a marked insulin resistance in GK rats with a major increment in the AUC and HOMA index (Figure 1D–F). The ITT, AUC, and HOMA were significantly reduced in GKL group (Figure 1D–F).

### 2.2. Endothelium-Dependent and Independent Vascular Relaxation

Endothelial-mediated vasodilation of phenylephrine-precontracted aortas in response to ACh was evaluated in the different groups with (+) or without (−) PVAT. Diabetic GK rats exhibited a marked endothelial dysfunction with normal endothelium-independent relaxations to SNP (Figure 2), as previously [6]. Maximal endothelial-mediated vascular relaxation in response to ACh significantly improved in luteolin-treated Wistar and GK rats (Figure 2A,B, Table 2). Importantly, luteolin treatment significantly recovered endothelium-dependent vascular relaxation in arteries mounted with PVAT (Figure 2A–C, Table 2). In addition, vascular sensitivity to ACh was significantly improved in aorta of Wistar and GK rats treated with luteolin mounted with PVAT (WL + PVAT, GKL + PVAT; Figure 2A–C, Table 2) when compared with W and GK rats. Vascular sensitivity to SNP was also significantly improved in aorta of Wistar and GK rats treated with luteolin (WL, GKL; Figure 2D–F, Table 2) when compared with the respective control rats, while no differences on maximal relaxation were observed (Figure 2D–F; Table 2). Preincubation of the arterial rings with L-NAME significantly decreased the vasodilator response to ACh in all groups (data not shown). EC_50_ values and maximal relaxations are displayed in Table 2.

### 2.3. Vascular Contraction in Response to Endothelin-1

Aortic rings from diabetic GK rats mounted with PVAT exhibited a similar response to endothelin-1 (ET1) when compared with arteries mounted without PVAT (Figure 3). In GK rats, arteries mounted with PVAT did not change maximal contraction in response to ET1 in contrast to nondiabetic Wistar rats (Figure 3A,B). Aortas of diabetic rats mounted without PVAT and treated with luteolin did not significantly change ET1 contraction responses (Figure 3B). The presence of PVAT in aortas of Wistar and GK rats treated with luteolin markedly reduced contraction to endothelin-1 demonstrating a potent anticontractile effect of thoracic periaortic PVAT in these groups (Figure 3A–C). There is an increment in the vascular sensitivity to ET1 and a significative reduction in maximal contraction of WL + PVAT and GKL + PVAT arteries (Figure 3D, Table 2) highlighting the anticontractile effect of thoracic periaortic in the animals treated with luteolin. Detailed data on maximal contractions and EC_50_ values are summarized in Table 2. To explore the role of potassium channels in PVAT anticontractile response in luteolin-treated rats, we used 4-aminopiridine (4-AP). The results showed that incubation of aortic rings with 1 mM 4-AP (a voltage-dependent potassium channel (Kv) inhibitor) diminished the relaxation effect observed in the presence of PVAT in WL + PVAT and GKL + PVAT groups (Figure 3D), indicating Kv involvement in the anticontractile effect observed in arteries mounted with PVAT.

### 2.4. Oxidative Stress and Glycation in the Vascular Wall

We determined the levels of superoxide anion in the diabetic vasculature. Diabetic aortic rings of GK rats showed a 2-fold increment in superoxide production, in agreement with previous results from our group [19]. Moreover, diabetic GK rats displayed enhanced immunoreactive nitrotyrosine and advanced glycation end-products’ levels in their aortas (Figure 4). Luteolin treatment significantly decreased vascular superoxide anion (Figure 4A,B), nitrotyrosine (Figure 4C,D), and AGEs’ levels in diabetic GK rats (Figure 4E,F).

### 2.5. PVAT Oxidative and Inflammatory Biomarkers

Previous work from our laboratory revealed that PVAT presents reduced antioxidant defenses and increased oxidative stress in diabetic GK rats [6]. To evaluate the impact of luteolin in the antioxidant features of periaortic adipose tissue, we evaluated the antioxidant enzyme activities of Mn-superoxide dismutase (MnSOD) and AR. Periaortic adipose tissue of GK rats has a significant reduction in the activity of MnSOD (Figure 5A) while the activity of AR was significantly higher when compared with Wistar controls (Figure 5B). Luteolin treatment significantly incremented MnSOD activity and decreased AR activity in PVAT of GK rats (Figure 5A,B). Periaortic adipose tissue of GK rats also exhibits lower levels of GSH and higher levels of malonaldehyde (Figure 5C,D) relative to age-matched Wistar rats. Luteolin treatment significantly decreased MDA levels and recovered GSH levels (Figure 5C,D).

Inflammatory biomarkers and nitrotyrosine levels are elevated in PVAT of GK rats highlighting the pro-inflammatory profile in periaortic adipose tissue from diabetic rats [6]. Herein, we demonstrate that luteolin treatment significantly decreased periaortic adipose tissue CRP and CCL2 levels (Figure 6A,B), in addition with a decrement in PVAT nitrotyrosine levels (Figure 6C,D) reducing PVAT pro-inflammatory and pro-oxidant profile (Figure 6A–D).

## 3. Discussion

Experimental and in vitro studies suggest that luteolin can have antioxidant and anti-inflammatory properties [20,21]. In contrast, no study has investigated the impact of luteolin treatment in the modulation of PVAT properties and its consequences on vascular dysfunction in T2D. Therefore, the impact of luteolin treatment was evaluated in vivo on vascular function and periaortic adipose tissue of diabetic GK rats providing new insights to understand the mechanisms of action underlying this nutraceutical. Herein, we reported for the first time, that luteolin exerts its beneficial endothelial effects in an animal model of T2D, in part, through inhibition of vascular oxidative stress and glycation. Importantly, luteolin accumulates in PVAT and reduces inflammation and oxidative stress in this adipose tissue rescuing its anticontractile phenotype and contributing to an improvement in endothelial function.

Luteolin plays a key role in the maintenance of insulin sensitivity and overall metabolism explaining the global benefits on glucose and lipid metabolism observed in this study. It was previously reported that luteolin was able to reduce hepatic steatosis and insulin resistance [15,16,17] in mice with diet-induced obesity [16]. In addition, it was previously shown that luteolin facilitates insulin action in adipocytes by activating the peroxisome proliferator-activated receptor (PPAR-γ) pathway [22]. Luteolin treatment for 24 h reportedly increases the response of glucose uptake to insulin stimulation in 3T3-L1 adipocytes enhancing Akt2 phosphorylation in an insulin-stimulated state [22]. Herein, we show that insulin resistance in GK rats is significantly reduced by luteolin treatment, highlighting the major role of this flavone in controlling insulin sensitivity. In agreement, previous studies have reported that luteolin decreases insulin resistance in adipose tissue macrophages through AMP-activated protein kinase α1 signaling, and due to an attenuation in inflammatory responses both in macrophages [23] and adipocytes through repression of nuclear factor-κB [24].

Luteolin can act as a naturally-dipeptidyl peptidase IV inhibitor and exert protective and beneficial effects on insulin secretion from β-cells [25,26], explaining the anti-diabetic properties of this flavone. In addition, luteolin has been reported to inhibit alpha glucosidase [27], justifying the reduction of blood glucose observed in our study. Importantly, luteolin significantly improves glucose levels (fasting and after IPGTT) in diabetic GK rats. Accordingly, several flavonoids were described to reduce glucose levels in experimental diabetes [28,29]. Flavonoids have a comparable lipophilic backbone to PPAR-γ ligands. In addition, some flavonoids have beneficial pharmacological actions on carbohydrates and lipid metabolism via modulation of PPAR-γ [30,31,32]. Specifically, flavonoids of flavones groups such as luteolin were able to activate PPAR-γ receptor directly [30].

Noteworthy, luteolin significantly reduced body weight, adiposity index, and total cholesterol levels. In agreement, it was previously reported that luteolin significantly reduced body weight, liver and white adipose tissues weights, and elevated plasma lipids in obese mice [16,21]. Indeed, luteolin was able to inhibit liver X receptor activation [31], explaining the lipid-lowering effects observed.

Ex vivo studies in isolated aortic rings from control animal models have shown that luteolin affects vascular reactivity in an endothelium-dependent manner [11,32]. Herein, we show that luteolin treatment reverted ACh-induced relaxation of aortic rings mounted with PVAT in GK rats, improving endothelial function in this animal model of T2D. Luteolin increased SNP sensitivity, indicating that it also has endothelial-independent effects. In addition, luteolin resulted in an attenuation of the ET1-induced constriction response in aortas of W and diabetic GK rats mounted with PVAT. 4-AP (a Kv inhibitor) diminished this anticontractile effect observed in the presence of PVAT in WL + PVAT and GKL + PVAT, and reduced the relaxation potency indicating voltage-dependent potassium channel involvement. Accordingly, luteolin decreased high glucose-induced endothelial dysfunction through reduction of oxidative stress. Indeed, luteolin significantly reduced the oxidative stress due to exacerbated reactive oxygen species (ROS) levels and hydroxyl radical (OH) formation, and the decrement in NO level, NOS, and superoxide dismutase activity caused by high glucose [11]. In addition, luteolin induced vasorelaxation in the thoracic aorta in rats in an endothelial-independent fashion [32], apparently through the inhibition of sarcolemmal Ca^2+^ channels, released intracellular Ca^2+^ stores, and activated K^+^ channels [32].

Reduced NO bioavailability due to oxidative stress is the hallmark in the development of diabetic-induced endothelial dysfunction [4]. In T2D, vascular oxidative stress is significantly incremented, contributing to reduce NO bioavailability, endothelial dysfunction, and atherogenesis [6]. Herein, supplementation of luteolin significantly reduced vascular oxidative stress, confirming the previous reported antioxidant properties of this flavonoid [11,33]. Previous studies have shown that luteolin has protective effects in endothelial cells through eNOS activation and NO production [12,24]. In addition, previous studies have described that luteolin had cardioprotective effects on ischemia/reperfusion injury in diabetic rats [34], and protected rat mesenteric arteries from superoxide anion-induced injury [35]. Moreover, luteolin stopped the decrease of eNOS expression promoted by palmitic acid in human umbilical vein endothelial cells [36].

Previous studies unveiled the promising properties of some medicinal plants and its bioactive constituents (such as luteolin) in the management of advanced glycation end- product (AGE)-mediated vascular complications observed in diabetes and other metabolic disorders [37,38,39]. Herein, we show that luteolin was able to significantly reduce immunoreactive AGEs’ levels in the vascular wall, highlighting the potential of this compound to reduce oxidative stress, glycation, and vascular complications in diabetic conditions. Accordingly, previous studies described that luteolin exhibits carbonyl trapping properties and hence protects against AGEs in vitro [39]. This study reports, for the first time, this potential in type 2 diabetic animal models.

Luteolin attenuated vascular inflammation by suppressing tumor necrosis factor (TNF)-stimulated expression of chemokines and adhesion molecules, as well as preventing the activation of nuclear factor (NF)-κB signaling in experimental models [13]. Moreover, luteolin was also able to reduce mRNA expression of pro-inflammatory cytokines such as TNF and interleukin (IL)-6 in palmitic acid-induced IkB kinase (IKKb)/NF-κB activation in human umbilical vein endothelial cells [35]. In addition, luteolin inhibits chronic low-grade inflammation suppressing adipocyte-dependent activation of macrophages due to reduced JNK activation [40]. Furthermore, luteolin treatment was shown to reduce mRNA levels of TNFα, IL-6, and CCL2, while it enhanced the gene expression of adiponectin and leptin in 3T3-L1 adipocytes and primary mouse adipose cells. Most interestingly, the treatment with luteolin markedly increased PPARγ transcriptional activity in 3T3-L1 adipocytes, and luteolin-increased expression of adiponectin and leptin was abolished by GW9662, a PPARγ antagonist [22].

Importantly, inflammation of PVAT exerts harmful effects on the vasculature through the accumulation of M1 macrophages that produce and release several factors including cytokines [41]. In addition, visceral adipose tissue through pro-inflammatory cytokines reaching the portal circulation, induce liver and chronic inflammatory response leading to endothelial dysfunction. Luteolin was previously described with antioxidant and anti-inflammatory properties [20,21,40]. Our study shows that luteolin treatment was able to improve the PVAT phenotype, significantly reducing periaortic CRP and CCL2 levels while improving oxidative stress through an increment in antioxidant defense systems of diabetic GK rats.

The beneficial effects of luteolin may also be partly due to modulating effects on endogenous fat synthesis, and leptin-adiponectin formation. Indeed, the luteolin effects observed herein might be influenced by changes in adipokines such as leptin and adiponectin in adipocytes that exert effects on the PPARγ pathway [42,43]. Moreover, leptin and adiponectin exhibit vasoactive properties, which might also be having an impact on the results observed, considering the dual effects of leptin on blood pressure control [44,45].

The pleiotropic actions of luteolin as an anti-diabetic, antioxidant and anti-inflammatory nutraceutical with lipid-lowering effects and vascular function benefits strongly indicate that luteolin is a promising therapeutic agent to treat human vascular complications associated with metabolic diseases, and should be further investigated with clinical trials.

## 4. Materials and Methods

### 4.1. Animals

Animal care and all experimental procedures were done in accordance with the Portuguese Law on Experimentation with Laboratory Animals, based on the principles of laboratory animal care, as adopted by the Directive 2010/63/EU for all animal-based experiments. Animal studies were approved by the Animal Welfare Committee of University of Coimbra (ORBEA 25/2015) according to ARRIVE guidelines. Adult male Wistar (W) and Goto-Kakizaki (GK) rats were originated from the breeding colony in the Faculty of Medicine of the University of Coimbra (Portugal).

All animals were maintained in rooms with periods of light and darkness of 12 h each, with free access to water. Animals were fed with a standard commercial pellet chow (Diet AO3-Panlab). Individual housing and environmental enrichment were used.

The study was randomized and blinded. W and GK rats were randomized into two groups as follows: control groups treated with vehicle (W (*n* = 10) and GK (*n* = 10)), and experimental groups treated orally with luteolin (10 mg/kg/day) for 2 months (WL (*n* = 10) and GKL (*n* = 10)). Randomization was done by block of 4 animals per day with 2 animals allocated to the 2 groups: luteolin or control (GraphPad online calculator QuickCalcs—http://www.graphpad.com/quickcalcs/randomize1.cfm, September 2018), and all experiments were performed without knowledge of the treatments (blinding). At 6-months-old, we have determined the amount of food ingested daily, and administered luteolin (10 mg/kg/day) or vehicle mixed in the diet (ensuring that all the pellets are consumed) for 2 months.

At 8-months-old, the sacrifice of the animals was performed according to the norms of welfare and principles of animal-based experiments. For anesthesia, ketamine/chlorpromazine (ketamine chloride (75 mg·kg^−1^, i.m., Parke-Davis, Ann Arbor, MI, USA), chlorpromazine chloride (2.65 mg·kg^−1^, i.m., Lab. Vitória, Portugal)) were used. W and GK had a body weight of 432 ± 7.9 and 375 ± 10.9, respectively.

At the end of the treatment, blood was collected, and aorta arteries and PVAT were removed and used as described in the methodologies below.

### 4.2. Determination of Metabolic Parameters

After a fasting period of 15 h, animals were anesthetized with ketamine/chlorpromazine and sacrificed by cervical dislocation. Blood samples were taken by cardiac puncture to determine lipid levels.

Glucose tolerance tests were made using animals after overnight fasting. Blood glucose was determined by a glucose-oxidase method using a glucometer (Glucometer-Elite-Bayer S.A, Lisbon, Portugal) and compatible reactive test strips, by sampling from the tail vein at 0, 60, and 120 min, after injection with glucose (i.p. 1.75 g·kg^−1^) in PBS.

For insulin tolerance test, insulin (i.p. 0.25 U·mL^−1^·kg^−1^) was injected and glucose evaluated at 0, 15, 30, 45, 60, and 120 min. The insulin resistance test was done with animals in fasting state; the homeostasis model assessment of insulin resistance (HOMA) was calculated as described in [46]. HOMA was calculated as ((G0) x (I0)/22.5) where G0 is the fasting glucose level (mmol·L^−1^), and I0 is the fasting insulin level (mU·mL^−1^).

Lipids such as total cholesterol and triglycerides were measured, as previously [47,48], using commercially-available kits.

### 4.3. Isometric Tension Studies

Aortas were quickly removed and cleansed in modified Krebs-Henseleit buffer (composition in mM: NaCl 119; KCl 4.7; CaCl_2_ 1.6; MgSO_4_ 1.2; NaHCO_3_ 25; KH_2_PO_4_ 1.2; glucose 11.0). The aortas were cleaned, divided into two segments (one with, and one without PVAT), approximately 4 mm wide, and mounted in a wire myograph. Individual myograph chambers were filled with modified Krebs-Henseleit buffer (37 °C, pH 7.4) oxygenated (95% O_2_, 5% CO_2_). The aortic rings were subjected to a resting tension of 4.7 mN. After equilibrating for 60 min, all vessels were pre-constricted with 0.3 µM phenylephrine. Cumulative concentration-response curves to acetylcholine (ACh, 10^−9^ to 10^−2^ M), sodium nitroprusside (SNP, 10^−9^ to 10^−4^ M), phenylephrine or endothelin-1 were obtained as previously described [5,47,49].

### 4.4. Detection of Superoxide

Unfixed frozen, 6-µm-thick sections of aorta were incubated with DHE (2 × 10^−6^ M) in PBS for 30 min at 37 °C in a humified chamber protected from light. DHE oxides into ethidium bromide, upon reaction with O_2_^•−^, and binds to DNA in the nucleus originating red fluorescence [5]. To detect ethidium bromide, a fluorescence microscope was used (with a 568-nm filter; Leica DMIRE200, Wetzlar, Germany). Arteries were processed and imaged as previously [5,6]. Microscope and camera settings were kept constant for all preparations. ImageJ was used to quantify fluorescence (1.40 g, NIH).

### 4.5. Determination of Aortic Immunofluorescence

Aorta sections of 6 µm were washed in PBS and fixed in ice-cold acetone for 10 min. After that, sections were permeabilized in 1% Triton X-100/PBS (pH 7.4) for 10 min and blocked with 10% goat serum for 30 min. Primary antibodies (in PBS containing 0.02% BSA, PBS/BSA) were added and the sections were incubated at 4 °C overnight. Afterward, they were extensively washed with PBS/BSA solution and incubated with secondary antibodies for one hour. Aorta sections were washed, counterstained with 4′,6-diamidino-2-phenylindole, mounted, and visualized with Leica DMIRE200 fluorescence microscope as previously described [5,6].

### 4.6. Enzymatic Assays

Manganese superoxide dismutase (MnSOD) and aldose reductase (AR) activities were assessed as previously described [6].

### 4.7. Measurement of Glutathione Concentration

Glutathione (GSH) concentration was determined using a glutathione assay kit (Sigma, CS0260) and according to manufacturer’s instructions [6].

### 4.8. Statistical Analysis

All data were analyzed using GraphPad Prism PC Software (version 3.0, ANOVA, San Francisco, CA, USA) and are expressed as mean ± SE (*n* = 10 individual animals per group). ANOVA was used to evaluate statistical differences. *p* < 0.05 was considered significant. Dose-response curves were built-in by nonlinear regression (simplex algorithm) and compared with two-way ANOVA with subsequent Bonferroni post hoc test for individual comparison. The sample size was calculated with the software G*Power version 3.1.9.6. for effect size and confidence interval.

### 4.9. Materials

Antibodies against nitrotyrosine and anti-AGE (clone:6D12) were obtained from Merck Millipore (Darmstadt, Germany) and Trans Genic Inc. (Kumamoto, Japan), respectively. DHE was obtained from Invitrogen (Barcelona, Spain). CRP and CCL2 EIA kits were obtained from Cayman Europe (Tallinn, Estonia). All other reagents and chemicals utilized in the study were of high grade and were acquired from Sigma (St. Louis, MO, USA).

## 5. Conclusions

Our study shows that luteolin can improve systemic metabolic profile and vascular dysfunction associated with type 2 diabetes due to antioxidant and anti-inflammatory properties. Our study collectively demonstrates that luteolin has a potential application in the treatment of diabetes-related vascular diseases and may be a useful nutraceutical for some metabolic disorders.

## Figures and Tables

**Figure 1 ijms-22-13671-f001:**
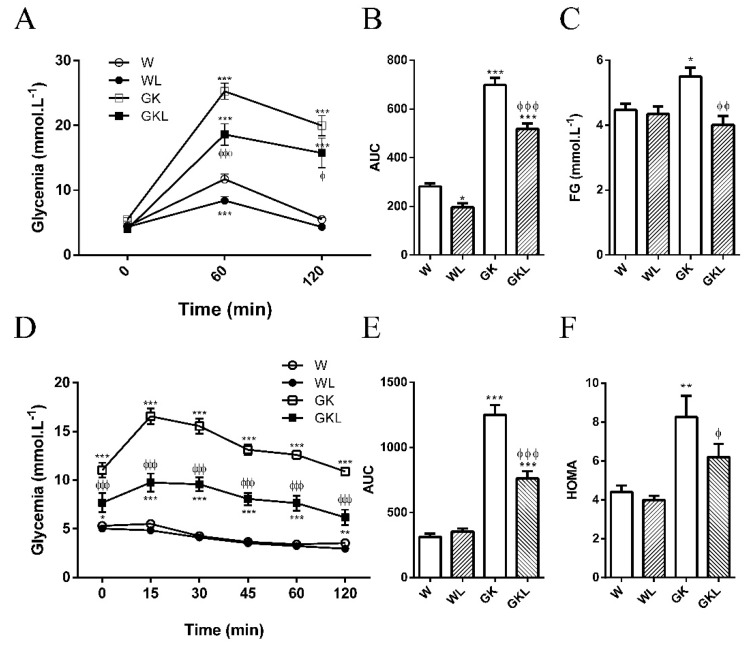
Influence of luteolin on blood glucose levels throughout an intraperitoneal glucose tolerance test (IPGTT; (**A**)), the glucose area under the curve (AUC; (**B**)), fasting glycemia (**C**), insulin tolerance test (ITT; **D**), the insulin area under the curve (AUC; (**E**)), and homeostasis model assessment of insulin resistance (HOMA, (**F**)) in normal Wistar (W), diabetic Goto-Kakizaki (GK) control rats and rats treated with luteolin (10 mg/kg/day, orally) for 2 months (WL, GKL). Data are expressed as mean ± SE (*n* = 10). * *p* < 0.05, ** *p* < 0.01, *** *p* < 0.001 vs. W group; ^ϕ^
*p* < 0.05, ^ϕϕ^
*p* < 0.01, ^ϕϕϕ^
*p* < 0.001 vs. GK group.

**Figure 2 ijms-22-13671-f002:**
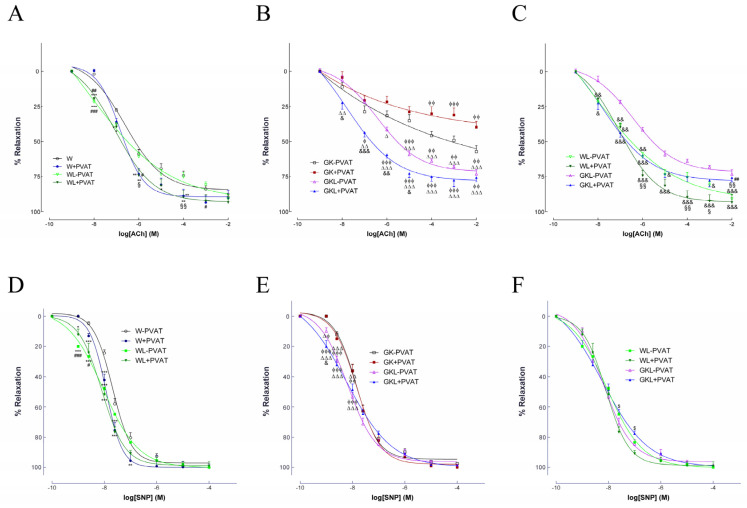
Effects of luteolin on vascular responses to acetylcholine (ACh, panel **A**–**C**), and sodium nitroprusside (SNP, panel **D**–**F**) in aortas of normal Wistar (W) and diabetic Goto-Kakizaki (GK) rats mounted with or without perivascular adipose tissue (PVAT). (**A**,**B**) Endothelium-dependent vasodilation in aortic rings in the absence (−PVAT) or presence of PVAT (+PVAT) was assessed in W (panel **A**) and GK rats (panel **B**) with or without luteolin treatment. (**C**) Comparison of endothelium-dependent vasodilation in aortic rings in W and GK rats treated with luteolin. (**D**). Vasodilator response to SNP in aortas of W (panel **D**) and GK (panel **E**) rats. Endothelium-independent vasodilation in aortic rings in the absence (−PVAT) or presence of PVAT (+PVAT) was assessed in W and GK with or without luteolin treatment (panel **D**–**F**). (**F**) Comparison of endothelium-independent vasodilation in aortic rings in W and GK rats treated with luteolin. Data are expressed as mean ± SE (*n* = 10). * *p* < 0.05, ** *p* < 0.01, *** *p* < 0.001 vs. W-PVAT group; ^#^
*p* < 0.05, ^##^
*p* < 0.01, ^###^
*p* < 0.001 vs. W + PVAT group; ^§^ *p* < 0.05, ^§§^ *p* < 0.01, ^§§§^ *p* < 0.001 vs. WL-PVAT group; ^$^ *p* < 0.05 vs. WL + PVAT group; ^ϕ^
*p* < 0.05, ^ϕϕ^
*p* < 0.01, ^ϕϕϕ^
*p* < 0.001 vs. GK-PVAT group; ^∆^ *p* < 0.05, ^∆∆^ *p* < 0.01, ^∆∆∆^ *p* < 0.001 vs. GK + PVAT group; ^&^ *p* < 0.05, ^&&^ *p* < 0.01, ^&&&^ *p* < 0.001 vs. GKL-PVAT.

**Figure 3 ijms-22-13671-f003:**
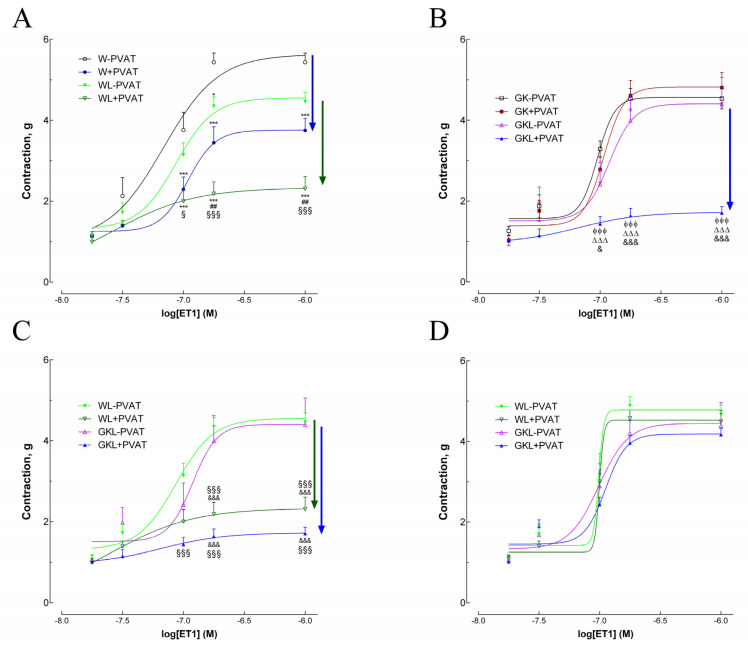
Effects of luteolin on contraction responses to endothelin-1 (ET1) in aortas of Wistar (W) and diabetic Goto-Kakizaki (GK) rats in the absence (−PVAT) or presence of PVAT (+PVAT). The effect of luteolin is highlighted in panel (**A**) and (**B**) for W and GK aortas, respectively. Luteolin recovered the anticontractile PVAT phenotype typical of normal W in diabetic GK rats (panel **B**,**C**). (**D**) Effects of luteolin on contraction responses to ET1 in aortas of W and GK rats in the presence of 4-aminopiridine, a voltage-dependent potassium channel (Kv) inhibitor. Luteolin treatment promoted a substantial PVAT anticontractile effect in aortic rings (panel **B**,**C**) lost after treatment with 4-aminopiridine (panel **D**). Data are expressed as mean ± SE (*n* = 10). * *p* < 0.05, *** *p* < 0.001 vs. W-PVAT group; ^##^ *p* < 0.01 vs. W + PVAT group; ^§^ *p* < 0.05, ^§§§^ *p* < 0.001 vs. WL-PVAT group; ^ϕϕϕ^
*p* < 0.001 vs. GK-PVAT group; ^∆∆∆^
*p* < 0.001 vs. GK + PVAT group; ^&^ *p* < 0.05, ^&&&^ *p* < 0.001 vs. GKL-PVAT.

**Figure 4 ijms-22-13671-f004:**
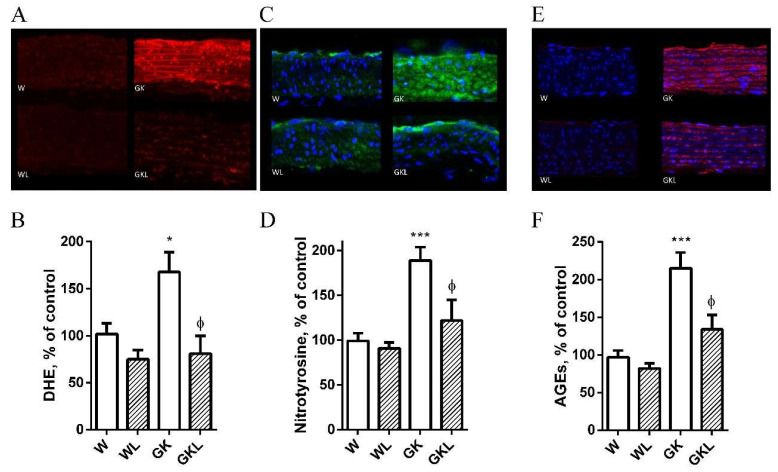
Effects of luteolin treatment on vascular oxidative stress and advanced glycation end- products’ levels in Wistar (W) and diabetic Goto-Kakizaki (GK) rats. Representative DHE-stained aorta artery sections reflect O_2_^•−^ production in W and GK rats with (WL, GKL) or without luteolin treatment (W, GK). (Panel **A**). The endothelium is facing up in all layers. At identical settings, fluorescence (reflecting O_2_^•−^ levels in the endothelium, intima, and media) in diabetic GK was markedly increased compared to age-matched control W rats. DHE fluorescence decreased in the diabetic GK rats treated with luteolin (GKL). Panel (**B**) contains quantification of the fluorescence ethidium signal in the different groups of arteries. Representative aortic sections showing nitrotyrosine staining (panel **C**) in W and GK rats with (WL, GKL) or without luteolin treatment (W, GK). Panel (**D**) contains quantification of the green fluorescence in the different groups of arteries. Representative aortic sections showing AGE staining (panel **E**) in W and GK rats with (WL, GKL) or without luteolin treatment (W, GK). Panel (**F**) exhibits quantification of the red fluorescence in the various groups of arteries. Data are expressed as mean ± SE (*n* = 10). * *p* < 0.05, *** *p* < 0.001 vs. W group; ^ϕ^
*p* < 0.05 vs. GK group.

**Figure 5 ijms-22-13671-f005:**
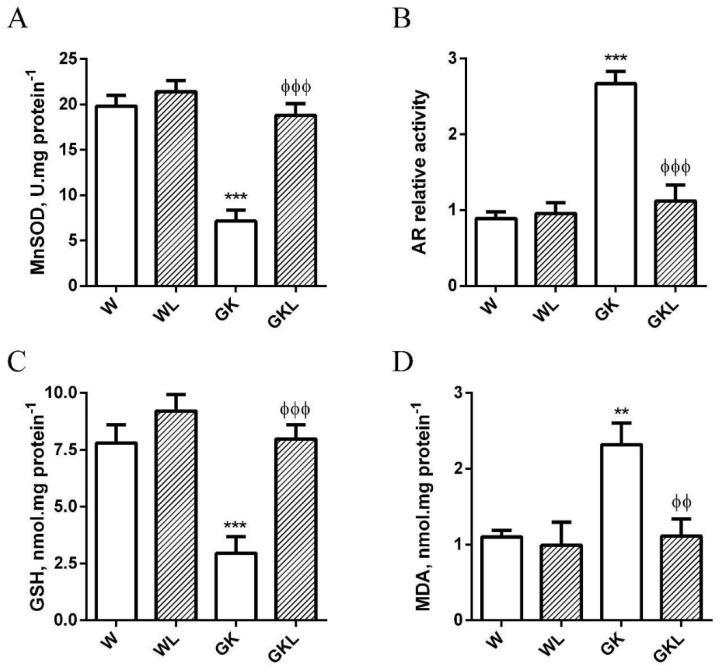
Effects of luteolin treatment on reduced antioxidant defenses and increased lipid peroxidation in perivascular adipose tissue (PVAT) of thoracic aorta from diabetic Goto-Kakizaki (GK) rats. Activities of manganese superoxide dismutase (MnSOD; **A**) and aldose reductase (AR; **B**), tissue contents of glutathione (GSH; **C**), and malonaldehyde levels (MDA; **D**) in PVAT of thoracic aortas of GK rats compared with nondiabetic Wistar (W) rats treated with (WL, GKL) or without luteolin. Data are expressed as mean ± SE (*n* = 10). *** *p* < 0.001 vs. W group; ^ϕϕϕ^
*p* < 0.001 vs. GK group; ** *p* < 0.01 vs. W group; ^ϕϕ^
*p* < 0.01, vs. GK group.

**Figure 6 ijms-22-13671-f006:**
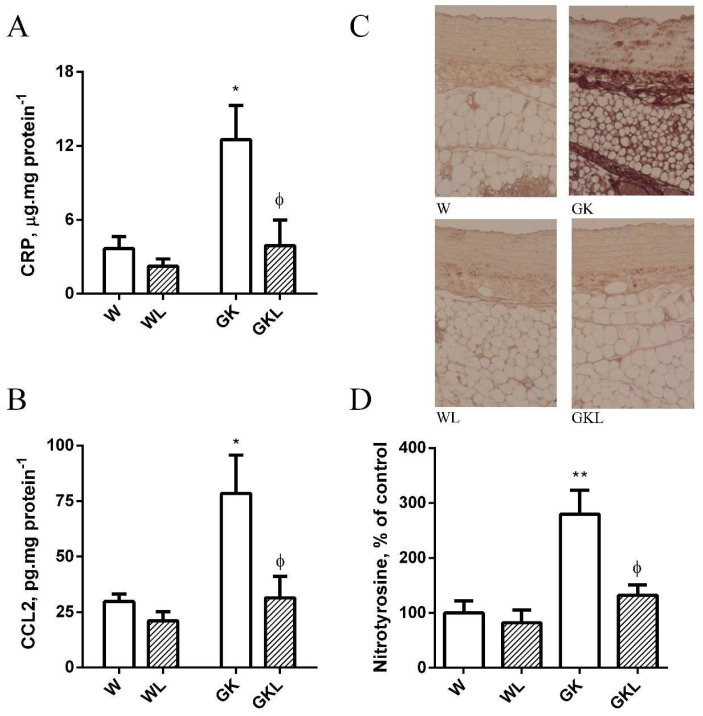
Effects of luteolin on inflammation of perivascular adipose tissue (PVAT) in thoracic aorta from Wistar and Goto-Kakizaki (GK) rats. CRP (**A**) and CCL2 (**B**) levels; immunohistochemical staining for nitrotyrosine (**C**,**D**) were determined in PVAT of thoracic aortas of the different groups of rats. Data are expressed as mean ± SE. * *p* < 0.05, ** *p* < 0.01 vs. W group; ^ϕ^
*p* < 0.05 vs. GK group.

**Table 1 ijms-22-13671-t001:** Body weight, adiposity index, triglycerides, and cholesterol (total and non-HDL) levels in nondiabetic Wistar (W) and diabetic Goto-Kakizaki (GK) rats with (WL, GKL) or without luteolin treatment (W, GK).

	W	WL	GK	GKL
BW (g)	432.1 ± 7.9	401.7 ± 5.3 ***	375 ± 10.9 ***	350.6 ± 7.9 ***^,^ ^ϕϕϕ^
Adiposity index (%)	2.77 ± 0.14	2.05 ± 0.12 ***	2.4 ± 0.09 ***	1.95 ± 0.11 ***^,^ ^ϕϕϕ^
Triglycerides (mmol·L^−1^)	0.6 ± 0.03	0.45 ± 0.06	1.07 ± 0.04 ***	1.21 ± 0.07 ***
Total cholesterol (mmol·L^−1^)	1.92 ± 0.05	1.91 ± 0.04	3.5 ± 0.06 ***	2.97 ± 0.1 ***^,^ ^ϕϕϕ^
Non-HDL cholesterol (mmol·L^−1^)	0.89 ± 0.04	0.8 ± 0.03 *	1.36 ± 0.04 ***	0.9 ± 0.04 ^ϕϕϕ^

Adiposity index = sum of weights of white adipose tissues divided by body weight × 100. Data are expressed as mean ± SE (*n* = 10 animals in each group). *** *p* < 0.001 vs. W rats; ^ϕϕϕ^
*p* < 0.001 vs. GK rats; * *p* < 0.05, vs. W rats.

**Table 2 ijms-22-13671-t002:** Maximal relaxation responses (%) and −logEC50 in isolated aorta arteries of diabetic Goto-Kakizaki (GK) rats and age-matched nondiabetic Wistar rats with or without perivascular adipose tissue (PVAT). W and GK rats treated with luteolin (WL, GKL) with (+PVAT) or without (−PVAT) perivascular adipose tissue. pEC50 values are presented as the negative logarithm (−logEC50) of concentration of the agonist.

	W − PVAT	W + PVAT	WL − PVAT	WL + PVAT	GK − PVAT	GK + PVAT	GKL − PVAT	GKL + PVAT
ACh								
pEC50	6.58 ± 0.1	6.78 ± 0.1	8.25 ± 0.5 ***^,###^	7.06 ± 0.2 **^,§§§^	6.5 ± 0.14 ^§§§,$$$^	6.3 ± 0.21 ^##,§§§,$$$^	6.43 ± 0.14 ^§§§,$$$^	7.78 ± 0.5 ***^,###,§§,$$$, ϕϕϕ, ∆∆∆,^ ^&&&^
Maximal relaxation (%)	86.3 ± 2.1	91.3 ± 2.3	91.8 ± 4.8 *	93 ± 2.8 **	53.2 ± 4.7 ***^,###,§§§,$$$^	35.4 ± 4.6 ***^,###,§§§,$$$,^ ^ϕϕϕ^	71.5 ± 3.9 ***^,###,§§§,$$$,^ ^ϕϕϕ, ∆∆∆^	78.0 ± 2.7 ***^,###,§§§,$$$,^ ^ϕϕϕ^^,^ ^∆∆∆,^ ^&&^
SNP								
pEC50	7.68 ± 0.03	7.91 ± 0.05 ***	8.05 ± 0.12 ***^,##^	8.1 ± 0.07 ***^,###^	7.76 ± 0.07 ^##,§§§,$$$^	7.78 ± 0.08 ^##,§§§,$$$^	8.2 ± 0.1 ***^,###,§§, ϕϕϕ, ∆∆∆^	8.2 ± 0.06 ***^,###,§§, ϕϕϕ^^,^ ^∆∆∆^
Maximal relaxation (%)	99.1 ± 1.2	96.9 ± 1.47	100.1 ± 2.9	99.7 ± 2.1	97.6 ± 3.3	99.5 ± 2.07	99.8 ± 3.7	96.3 ± 2.66
ET1								
pEC_50_	7. 16 ± 0.14	6.99 ±0.09 ***	7.06 ± 0.07	7.57 ± 0.1 ***^,###,§§§^	7.01 ± 0.03 **^,$$$^	6.99 ± 0.07 ***^,$$$^	6.97 ±0.04 ***^,§§,$$$, ϕ^	7.18 ± 0.05 ^###,§,$$$,^ ^ϕϕϕ^^,^ ^∆∆∆^^,&&&^
Maximal contraction (g)	5.43 ± 0.23	3.72 ± 0.23 ***	4.54 ± 0.25 ***^,###^	2.33 ± 0.3 ***^,###,§§§^	4.51 ± 0.25 ***^,###,$$$^	4.78 ± 0.4 ***^,###,$$$^	4.41 ± 0.2 ***^,###,$$$, ∆^	1.73 ± 0.17 ***^,###,§§§,$$,^ ^ϕϕϕ^^,^ ^∆∆∆^^,&&&^

Data are expressed as mean ± SE (*n* = 10 animals in each group). * *p* < 0.05, ** *p* < 0.01, *** *p* < 0.001 vs. W-PVAT group; ^##^ *p* < 0.01, ^###^ *p* < 0.001 vs. W + PVAT group; ^§^ *p* < 0.05, ^§§^ *p* < 0.01, ^§§§^ *p* < 0.001 vs. WL − PVAT group; ^$$^ *p* < 0.01, ^$$$^ *p* < 0.001 vs. WL + PVAT group; ^ϕ^
*p* < 0.05, ^ϕϕϕ^
*p* < 0.001 vs. GK-PVAT group; ^∆^ *p* < 0.05, ^∆∆∆^ *p* < 0.001 vs. GK + PVAT group; ^&&^ *p* < 0.01, ^&&&^ *p* < 0.001 vs. GKL-PVAT group.

## Data Availability

Data is contained within the article or Appendix A.

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
