# Peer review of "Luteolin Improves Perivascular Adipose Tissue Profile and Vascular Dysfunction in Goto-Kakizaki Rats"

_ijms, 2021, doi:10.3390/ijms222413671_

Round 1

Reviewer 1 Report

In general, the work is disorganized, poorly discussed, not all tests are discussed, the way it is presented makes it difficult to read and understand the findings.

  1. Obesity represents the most important risk factor in the pathogenesis of type 2 diabetes. Why choose a non-obese diabetic model of T2D?
  2. How to confirm that there is no significant change in food intake between groups (data not shown)? In addition, the body weight and adiposity index of GK rats are lower than Wistar rats; but TG is higher. What is the meaning?
  3. The age and weight of GK rats and Wistar rats should be described in the material method.
  4. Figure 2 is not clear enough. Please provide a higher magnification.
  5. What is the mechanism of luteolin to improve the perivascular adipose tissue profile and vascular dysfunction? This is worth exploring.

Author Response

Thank you very much for your valuable comments and suggestions.

In general, the work is disorganized, poorly discussed, not all tests are discussed, the way it is presented makes it difficult to read and understand the findings.

-Please note that the manuscript, in particular the discussion, was improved to meet the comments of the reviewers. The organization of discussion is as follows:

First paragraph – Novelty and overall achievements of this study.

Second paragraph – discussion of the effect of luteolin on insulin resistance

Third paragraph -- discussion of the effect of luteolin on glucose levels (the anti-diabetic properties)

4 paragraph – discussion of the effects on body weight, adiposity, and lipid profile

5 paragraph – discussion of the effects of luteolin on vascular function

6 paragraph – discussion of the effects of luteolin on oxidative stress

7 paragraph – discussion of the effects of luteolin on glycation

8 paragraph – discussion of the effects of luteolin on vascular inflammation

9 paragraph – discussion of the effects of luteolin on PVAT inflammation

10 paragraph – discussion of other potential mechanisms involved

11 paragraph- a broader view of potential underlying mechanisms and the potential translation to the clinic.

  1. Obesity represents the most important risk factor in the pathogenesis of type 2 diabetes. Why choose a non-obese diabetic model of T2D?

-Using a non-obese rat model such as Goto-kakizaki rats the authors can discriminate between the effects of diabetes (high glucose and insulin resistance) per se. These rats display mild hyperglycaemia, insulin resistance, vascular dysfunction and adipose tissue dysfunction making them a good model to evaluate pharmacological therapies (page 1, line 38-39).

  1. How to confirm that there is no significant change in food intake between groups (data not shown)? In addition, the body weight and adiposity index of GK rats are lower than Wistar rats; but TG is higher. What is the meaning?

-The food intake was monitored during luteolin treatment (weighed daily) and did not change during the experimental period between W/WL and GK/GKL, respectively (Figure 1supplement). This was clarified in the beginning of results (page 2, line 75-76).

GK rats have a lower body weight compared with Wistar rats (typical of this strain) but display polyphagia, polyuria, and polydipsia main features of T2D. In addition, due to a dysfunction in lipid metabolism, GK rats display high TG levels.

  1. The age and weight of GK rats and Wistar rats should be described in the material method.

-Please note that this was added as suggested (page 12, line 375 and line 378-379).

  1. Figure 2 is not clear enough. Please provide a higher magnification.

-Please note that it was provided as requested.

  1. What is the mechanism of luteolin to improve the perivascular adipose tissue profile and vascular dysfunction? This is worth exploring.

The authors present a mechanism that involves an antioxidant and anti-inflammatory effect of luteolin. “In addition, the beneficial effects of luteolin may also be partly due to modulating effects on endogenous fat synthesis, and leptin-adiponectin formation. Indeed, the luteolin effects observed herein might be influenced by changes in adipokines like leptin and adiponectin in adipocytes that exert effects on the PPARγ pathway [46, 47]. Moreover, leptin and adiponectin exhibit vasoactive properties, which might be also having an impact on the results observed considering the dual effects of leptin on blood pressure control [48, 49].”

Added to page 11, line 342-347.

Reviewer 2 Report

GENERAL COMMENTS

The manuscript addresses a topic of scientific interest, which is within the journal’s scope. Obesity and insulin resistance have been linked to a low-grade chronic inflammatory response characterized by increased macrophage infiltration, altered cytokine production and activation of inflammatory signalling pathways in adipose tissue. Pharmacological agents and natural products that are capable of reducing inflammatory activity and possess anti-diabetic properties are of interest in this regard. Luteolin, a naturally occurring flavonoid, has been demonstrated to inhibit lipopolysaccharide-induced tumor necrosis factor-α (TNFα) release and activation of NF-κB pathway in macrophages. However, less is known about the mechanisms and effects of luteolin on inflammation-related insulin resistance in adipocytes.  The authors have focused on the beneficial effects of luteolin on PVAS adipose tissue and vascular function in an animal model of Wistar (control) and (Goto-Kakizaki (nonobese-T2D) rats. A broader view of potential underlying mechanisms might be interesting to comment as well as the potential translation to the clinic.

The manuscript may benefit from considering the following aspects:

The abstract does not contain quantitative, statistical information. It would be good to know what is the magnitude of the effects observed.

Abstract, line 11: correct Kg by kg; Abstract, line 19: correct recued

Introduction, page 2, lines 54-63: formulate your specific hypothesis.

At the end of the paragraph delete the extra full stop.

As shown by Ding et al (referenced by the authors) luteolin treatment for 24 h reportedly increases the response of glucose uptake to insulin stimulation in 3T3-L1 adipocytes enhancing Akt2 phosphorylation in an insulin-stimulated state. Furthermore, luteolin treatment was shown to decrease mRNA levels of TNFα, interleukin-6 and MCP-1, while it increased the gene expression of adiponectin and leptin in 3T3-L1 adipocytes and primary mouse adipose cells. Most interestingly, the treatment with luteolin markedly enhanced PPARγ transcriptional activity in 3T3-L1 adipocytes, and luteolin-increased expression of adiponectin and leptin was blocked by GW9662, a PPARγ antagonist (Ding L, Jin D, Chen X. Luteolin enhances insulin sensitivity via activation of PPARγ transcriptional activity in adipocytes. J Nutr Biochem. 2010 Oct;21(10):941-7. doi: 10.1016/j.jnutbio.2009.07.009. Epub 2009 Dec 1. PMID: 19954946). From a mechanistic point of view, the beneficial effects may be partly due to modulating effects on endogenous fat synthesis, and leptin-adiponectin formation among others. In this regard, do the authors have data on leptin and adiponectin in their rats? If not, it would be necessary to comment in the Discussion that the luteolin effects observed might be influenced by changes in adipokines like leptin and adiponectin in adipocytes that exert effects on the PPARγ pathway (Muruzábal FJ, Frühbeck G, Gómez-Ambrosi J, Archanco M, Burrell MA. Immunocytochemical detection of leptin in non-mammalian vertebrate stomach. Gen Comp Endocrinol. 2002 Sep;128(2):149-52. doi: 10.1016/s0016-6480(02)00072-2. PMID: 12392688  //  Monika P, Geetha A. The modulating effect of Persea americana fruit extract on the level of expression of fatty acid synthase complex, lipoprotein lipase, fibroblast growth factor-21 and leptin--A biochemical study in rats subjected to experimental hyperlipidemia and obesity. Phytomedicine. 2015 Sep 15;22(10):939-45. doi: 10.1016/j.phymed.2015.07.001. Epub 2015 Jul 17. PMID: 26321743).

Moreover, leptin and adiponectin exhibit vasoactive properties, which might be also having an impact on the results observed taking into account the dual effects of leptin on blood pressure control (Agabiti-Rosei C, Paini A, De Ciuceis C, Withers S, Greenstein A, Heagerty AM, Rizzoni D. Modulation of Vascular Reactivity by Perivascular Adipose Tissue (PVAT). Curr Hypertens Rep. 2018 May 7;20(5):44. doi: 10.1007/s11906-018-0835-5. PMID: 29736674  //  Frühbeck G, Gómez-Ambrosi J. Modulation of the leptin-induced white adipose tissue lipolysis by nitric oxide. Cell Signal. 2001 Nov;13(11):827-33. doi: 10.1016/s0898-6568(01)00211-x. PMID: 11583918).

Materials & Methods:

Page 11, line 318 and 323: replace “Kg” by “kg”; page 12, line 339: correct “testes”

References: leave only one numeration for the references.

Author Response

Thank you very much for your valuable comments and suggestions.

GENERAL COMMENTS

The manuscript addresses a topic of scientific interest, which is within the journal’s scope. Obesity and insulin resistance have been linked to a low-grade chronic inflammatory response characterized by increased macrophage infiltration, altered cytokine production and activation of inflammatory signalling pathways in adipose tissue. Pharmacological agents and natural products that are capable of reducing inflammatory activity and possess anti-diabetic properties are of interest in this regard. Luteolin, a naturally occurring flavonoid, has been demonstrated to inhibit lipopolysaccharide-induced tumor necrosis factor-α (TNFα) release and activation of NF-κB pathway in macrophages. However, less is known about the mechanisms and effects of luteolin on inflammation-related insulin resistance in adipocytes. The authors have focused on the beneficial effects of luteolin on PVAS adipose tissue and vascular function in an animal model of Wistar (control) and (Goto-Kakizaki (nonobese-T2D) rats.

A broader view of potential underlying mechanisms might be interesting to comment as well as the potential translation to the clinic.

-It was performed as suggested. Added to Page 11, line 348-351 as follows:

“The pleiotropic actions of luteolin as an anti-diabetic, antioxidant and anti-inflammatory nutraceutical with lipid lowering effects and vascular function benefits strongly indicate that luteolin is a promising therapeutic agent to treat human vascular complications associated with metabolic diseases and should be further investigated with clinical trials.”

The manuscript may benefit from considering the following aspects:

The abstract does not contain quantitative, statistical information. It would be good to know what is the magnitude of the effects observed.

-Please considered that the quantification and statistical information were added in abstract.

Abstract, line 11: correct Kg by kg; Abstract, line 19: correct recued

--It was corrected as suggested.

Introduction, page 2, lines 61-63: formulate your specific hypothesis.

“We hypothesized that luteolin might ameliorate endothelial dysfunction and reduce PVAT pro-inflammatory and vasoconstrictive profile in diabetic GK rats.” Page 2, line 62-64.

At the end of the paragraph delete the extra full stop.

-- It was corrected as suggested.

As shown by Ding et al (referenced by the authors) luteolin treatment for 24 h reportedly increases the response of glucose uptake to insulin stimulation in 3T3-L1 adipocytes enhancing Akt2 phosphorylation in an insulin-stimulated state. Furthermore, luteolin treatment was shown to decrease mRNA levels of TNFα, interleukin-6 and MCP-1, while it increased the gene expression of adiponectin and leptin in 3T3-L1 adipocytes and primary mouse adipose cells. Most interestingly, the treatment with luteolin markedly enhanced PPARγ transcriptional activity in 3T3-L1 adipocytes, and luteolin-increased expression of adiponectin and leptin was blocked by GW9662, a PPARγ antagonist (Ding L, Jin D, Chen X. Luteolin enhances insulin sensitivity via activation of PPARγ transcriptional activity in adipocytes. J Nutr Biochem. 2010 Oct;21(10):941-7. doi: 10.1016/j.jnutbio.2009.07.009. Epub 2009 Dec 1. PMID: 19954946).

From a mechanistic point of view, the beneficial effects may be partly due to modulating effects on endogenous fat synthesis, and leptin-adiponectin formation among others. In this regard, do the authors have data on leptin and adiponectin in their rats?

--Unfortunately, no data regarding leptin and adiponectin are available.

If not, it would be necessary to comment in the Discussion that the luteolin effects observed might be influenced by changes in adipokines like leptin and adiponectin in adipocytes that exert effects on the PPARγ pathway (Muruzábal FJ, Frühbeck G, Gómez-Ambrosi J, Archanco M, Burrell MA. Immunocytochemical detection of leptin in non-mammalian vertebrate stomach. Gen Comp Endocrinol. 2002 Sep;128(2):149-52. doi: 10.1016/s0016-6480(02)00072-2. PMID: 12392688  //  Monika P, Geetha A. The modulating effect of Persea americana fruit extract on the level of expression of fatty acid synthase complex, lipoprotein lipase, fibroblast growth factor-21 and leptin--A biochemical study in rats subjected to experimental hyperlipidemia and obesity. Phytomedicine. 2015 Sep 15;22(10):939-45. doi: 10.1016/j.phymed.2015.07.001. Epub 2015 Jul 17. PMID: 26321743).

Moreover, leptin and adiponectin exhibit vasoactive properties, which might be also having an impact on the results observed considering the dual effects of leptin on blood pressure control (Agabiti-Rosei C, Paini A, De Ciuceis C, Withers S, Greenstein A, Heagerty AM, Rizzoni D. Modulation of Vascular Reactivity by Perivascular Adipose Tissue (PVAT). Curr Hypertens Rep. 2018 May 7;20(5):44. doi: 10.1007/s11906-018-0835-5. Frühbeck G, Gómez-Ambrosi J. Modulation of the leptin-induced white adipose tissue lipolysis by nitric oxide. Cell Signal. 2001 Nov;13(11):827-33. doi: 10.1016/s0898-6568(01)00211-x.

-Please note that the discussion was changed to address the valuable comments made by the reviewer. Suggestions are added (page 11, line 342-347) as follows:

“The beneficial effects of luteolin may be partly due to modulating effects on endogenous fat synthesis, and leptin-adiponectin formation. Indeed, the luteolin effects observed herein might be influenced by changes in adipokines like leptin and adiponectin in adipocytes that exert effects on the PPARγ pathway [46, 47]. Moreover, leptin and adiponectin exhibit vasoactive properties, which might be also having an impact on the results observed considering the dual effects of leptin on blood pressure control [48, 49].”

Materials & Methods:

Page 11, line 318 and 323: replace “Kg” by “kg”; page 12, line 339: correct “testes”

-- It was corrected as suggested.

References: leave only one numeration for the references.

-- It was corrected as suggested.
